# Numerical Scheme Based on the Implicit Runge-Kutta Method and Spectral Method for Calculating Nonlinear Hyperbolic Evolution Equations

**Yasuhiro Takei** [1,†] **and Yoritaka Iwata** [2,*,†]

1   Mizuho Research & Technologies, 2-3 Kanda-Nishiki-Cho, Chiyoda, Tokyo 101-8443, Japan; takei32yasuhiro@gmail.com
2   Department of Chemistry and Materials Engineering, Kansai University, 3-3-35 Yamate-Cho, Suita, Osaka 564-8680, Japan
*   Correspondence: iwata_phys@08.alumni.u-tokyo.ac.jp
†   These authors contributed equally to this work.

**Abstract:** A numerical scheme for nonlinear hyperbolic evolution equations is made based on the implicit Runge-Kutta method and the Fourier spectral method. The detailed discretization processes are discussed in the case of one-dimensional Klein-Gordon equations. In conclusion, a numerical scheme with third-order accuracy is presented. The order of total calculation cost is $O(N \log_2 N)$. As a benchmark, the relations between numerical accuracy and discretization unit size and that between the stability of calculation and discretization unit size are demonstrated for both linear and nonlinear cases.

**Keywords:** implicit Runge-Kutta method; Fourier spectral method; high-precision numerical scheme

**MSC:** 65M70; 65M12





## 1. Introduction

The dynamics of nonlinear hyperbolic equations are fascinating enough to be applicable to wave propagation on any scale, from elementary particles to waves on a cosmic scale. Even fundamental properties have not been fully understood for nonlinear hyperbolic problems, and it is difficult to elucidate these properties only by pure mathematical analysis. To understand the fundamental properties of nonlinear waves we employ numerical calculations. While in terms of treating nonlinear problems (e.g., various types of boundary, discontinuity such as shock propagation) it is important to individually specialize numerical schemes, we are going to establish a basic framework for calculating nonlinear hyperbolic evolution equations. In this context, much attention is paid to "universal applicability" and "reliability". In this paper, as benchmarks for linear and nonlinear hyperbolic problems, we start with reproducing solutions with a simple and general framework (i.e., spatially-continuous solutions under periodic boundary conditions).

We consider nonlinear hyperbolic evolution equations. For concrete examples of hyperbolic evolution equations, here we take one-dimensional linear and nonlinear Klein-Gordon equations (for a textbook, see [1]). The initial and boundary values problem of one-dimensional Klein-Gordon equations is written by

$$\begin{cases} \frac{\partial^2 u}{\partial t^2} + \alpha \frac{\partial^2 u}{\partial x^2} + \beta F(u) = 0, \\ u(x,0) = f(x), \ \ u(0,t) = u(L,t), \\ \frac{\partial u}{\partial t}(x,0) = g(x), \ \ \frac{\partial u}{\partial t}(0,t) = \frac{\partial u}{\partial t}(L,t), \end{cases} \tag{1}$$

for $(x, t) \in [0, L] \times [0, T]$, where $\alpha$, $\beta$ and $T$ are real numbers, and $f(x)$ and $g(x)$ are initial functions. The periodic boundary condition is imposed. Inhomogeneous term $F(u)$ is either linear or nonlinear function of $u$ (e.g., the polynomials and trigonometric functions of $u$). For constructing the numerical scheme, the equations are represented by the first-order evolution equations as for time.

$$\begin{cases} \frac{\partial u}{\partial t} = v, \\[2mm] \frac{\partial v}{\partial t} + \alpha \frac{\partial^2 u}{\partial x^2} + \beta F(u) = 0, \\[2mm] u(x, 0) = f(x), \quad u(0, t) = u(L, t), \\[2mm] v(x, 0) = g(x), \quad v(0, t) = v(L, t). \end{cases} \tag{2}$$

Various numerical schemes have been investigated to accurately and efficiently reproduce the nonlinear solutions of partial differential equations such as the nonlinear Klein-Gordon equations mentioned above. For nonlinear hyperbolic equations, numerical schemes well reproducing the conservation laws are required to be highly accurate, since the smoothing effect particularly associated with parabolic partial differential equations cannot be expected.

For the spatial discretization, a conventional finite difference method can be used to discretize the spatial variables in calculating Klein-Gordon equations, but it requires a very small spatial unit $\Delta x$ to keep sufficient accuracy. Furthermore, the problem of numerical dispersion is inevitable in typical finite difference methods in which numerical solutions are known to be difficult to satisfy the conservation laws with certain sufficient accuracy. Meanwhile, in the spectral method, the solution in the wavenumber space is known to be efficient to avoid the problem arising from the numerical dispersion, and rather easy to satisfy conservation laws to certain satisfactory degrees. Since the calculation cost of the spectral method is generally higher than that of the finite difference method, some approximation methods have been used to improve the feasibility of calculations (cf. pseudo-spectral method [2–5], collocation method [6,7]). Note that the spectral method is a discretization method in which the solution is represented by a linear combination of a finite number of Fourier series, and therefore the boundary condition is a periodic boundary condition. On the other hand, numerical schemes (e.g., spectral element method, spectral penalty method, etc. [8–11]) have been also developed for various boundary conditions such as Dirichlet boundary condition, Neumann boundary condition, and other boundary conditions.

For the time discretization, explicit methods have been used in calculating Klein-Gordon equations, but it requires additional treatments to obtain sufficiently accurate solutions. While linear and nonlinear solvers based on explicit methods are relatively simple with low computation cost, it is also known that there is a restriction called the Courant-Friedrichs-Lewy Condition (CFL condition, for short) on the time unit $\Delta t$ in order to obtain numerical results stably. Therefore, a numerical scheme combining explicit and implicit methods has been studied: e.g., a method for weakening the restriction on $\Delta t$ with keeping the stability of calculations [12–14]. In this context unconditionally-stable fully implicit method has been studied recently [15–17]. When it comes to a fully implicit method, the problem is generally reduced to a nonlinear type self-consistent equation, and it is necessary to apply numerical iteration. Numerical iterative methods generally require extra-ordinary high computational costs compared to the explicit method (non-iterative methods). The convergence and efficient implementation of numerical iterative methods have also come into the recent spotlight.

In this paper, we discuss how to construct a high-precision scheme for hyperbolic evolution equations. Based on the two-stage and third-order implicit Runge-Kutta method [18] and the spectral method [2,4,19–21], the order of computation is confirmed to be $O(N \log_2 N)$ in the benchmark calculations. This order estimate simply shows the feasibility and the

applicability of the proposed scheme. Consequently, the precision and stability of the numerical scheme are examined for linear and nonlinear cases. As a remark for the limitation of the proposed method, we will not go into a detailed discussion of boundary conditions other than the periodic boundary conditions. However, by incorporating spectral elements methods, spectral penalty methods, and so on, it is definitely possible to construct numerical schemes for the other boundary conditions.

## 2. Discretization of Space Using Spectral Method

The spectral method [2,4,19–21] is employed to discretize the spatial variables. Let the solution of Equation (2) be expanded by the Fourier series.

$$
\begin{cases}
u(x,t) = a_0(t) + \sum_{k=1}^{N} a_k(t) \cos\left(\frac{2\pi}{L}kx\right) + \sum_{k=1}^{N} b_k(t) \sin\left(\frac{2\pi}{L}kx\right), \\[2mm]
v(x,t) = c_0(t) + \sum_{k=1}^{N} c_k(t) \cos\left(\frac{2\pi}{L}kx\right) + \sum_{k=1}^{N} d_k(t) \sin\left(\frac{2\pi}{L}kx\right).
\end{cases}
\tag{3}
$$

Then substitute them into the first equation of (2). After multiplying $\cos\left(\frac{2\pi}{L}lx\right)$ and $\sin\left(\frac{2\pi}{L}lx\right)$ respectively, they are integrated for $\Omega = [0, L]$ with respect to $x$. It follows that

$$
\begin{cases}
\frac{da_0}{dt} = c_0, \\[2mm]
\frac{da_l}{dt} = c_l, \quad (l = 1, \cdots, N), \\[2mm]
\frac{db_l}{dt} = d_l, \quad (l = 1, \cdots, N).
\end{cases}
\tag{4}
$$

Similarly, substitute Equation (3) into the second equation of (2). After multiplying $\cos\left(\frac{2\pi}{L}lx\right)$ and $\sin\left(\frac{2\pi}{L}lx\right)$ respectively, they are integrate for $\Omega = [0, L]$ with respect to $x$. It follows that

$$
\begin{cases}
L\frac{dc_0}{dt} + \beta \int_0^L F(u)dx = 0, \\[2mm]
\frac{L}{2}\frac{dc_l}{dt} - \left(\alpha\frac{2\pi^2}{L}\right)l^2 a_l + \beta \int_0^L F(u) \cos\left(\frac{2\pi}{L}lx\right)dx = 0, \ (l = 1, \cdots, N), \\[2mm]
\frac{L}{2}\frac{dd_l}{dt} - \left(\alpha\frac{2\pi^2}{L}\right)l^2 b_l + \beta \int_0^L F(u) \sin\left(\frac{2\pi}{L}lx\right)dx = 0, \ (l = 1, \cdots, N).
\end{cases}
\tag{5}
$$

The solution to the original Equation (2) is obtained by solving Equations (4) and (5) in which $a_0$, $c_0$ and $a_l$, $b_l$, $c_l$, $d_l$ $(l = 1, \cdots, N)$ are found.

In terms of dealing with the nonlinearity, the following integral values appearing in Equation (5) is the bottle neck of the computational cost.

- $\int_0^L F(u)dx$
- $\int_0^L F(u) \cos\left(\frac{2\pi}{L}lx\right)dx$
- $\int_0^L F(u) \sin\left(\frac{2\pi}{L}lx\right)dx$

To deal with these terms, we use the operator transformation method [2,4,19,20]. Its main idea is not to solve the nonlinear term in the Fourier-transformed momentum space, but in the original coordinate space by carrying out the inverse Fourier transform. This procedure remarkably reduce the computational cost; indeed, if we introduce the approximation based on the trapezoidal formula, the nonlinear terms are written by

$$
\begin{cases}
\int_0^L F(u) \cos\left(\frac{2\pi}{L}lx\right)dx \simeq \frac{L}{J} \sum_{j=0}^{J-1} F(u_j) \cos\left(\frac{2\pi}{L}lx_j\right), \\[2mm]
\int_0^L F(u) \sin\left(\frac{2\pi}{L}lx\right)dx \simeq \frac{L}{J} \sum_{j=0}^{J-1} F(u_j) \sin\left(\frac{2\pi}{L}lx_j\right), \\[2mm]
\int_0^L F(u)dx \simeq \frac{L}{J} \sum_{j=0}^{J-1} F(u_j),
\end{cases}
\tag{6}
$$

where, under the periodic boundary condition, the equidistant $J$ division of $\Omega$ is denoted as $x_j$ for $j = 0, \cdots, J$, and the value of $u$ in time $t$ at each point is denoted as $u_j = u(x_j, t)$. Here, the right-hand side of the above equation is characterized by the fact that it is expressed in the same form as the discrete Fourier transform. Therefore, when calculating the integral (6) numerically, $u_j$ is obtained by the discrete inverse Fourier transform from $a_0(t)$, $a_l(t)$, and $b_l(t)$, $(l = 1, \cdots, N)$, and the right side of Equation (6) is obtained by the discrete Fourier transform from $F(u_j)$, $(j = 0, \cdots, J-1)$. Note that if $F(u)$ is a $M$-degree polynomial of $u$, the values of the left and right sides coincide for $J \geq (M+1)N + 1$ [2,4,19,20]. Furthermore, by using the Fast Fourier Transform (FFT) for the above discrete inverse Fourier transform and discrete Fourier transform, the computational cost of (6) becomes $O(N \log_2 N)$. Consequently, the spatially-discretized equation becomes

$$
\begin{cases}
\frac{da_0}{dt} = c_0, \\[2mm]
\frac{da_l}{dt} = c_l, \\[2mm]
\frac{db_l}{dt} = d_l, \\[2mm]
\frac{dc_0}{dt} = -\frac{\beta}{J} \sum_{j=0}^{J-1} F(u_j), \\[2mm]
\frac{dc_l}{dt} = \alpha \left(\frac{2\pi l}{L}\right)^2 a_l - \frac{2\beta}{J} \sum_{j=0}^{J-1} F(u_j) \cos\left(\frac{2\pi}{L} l x_j\right), \\[2mm]
\frac{dd_l}{dt} = \alpha \left(\frac{2\pi l}{L}\right)^2 b_l - \frac{2\beta}{J} \sum_{j=0}^{J-1} F(u_j) \sin\left(\frac{2\pi}{L} l x_j\right),
\end{cases}
\tag{7}
$$

based on the spectral method with the operator transformation treatment. For the discretization of spatial variables by the spectral method, the pseudo-spectral method and collocation methods are sometimes applied as effective numerical solution methods [2–7]. In particular, integral values are approximately obtained by weakening the condition in which the number of fractional points is taken to be $J \geq (M+1)N + 1$. In such situations, there is a risk that the conservation law may not be well reproduced due to an aliasing error arising from the overlaps between different wave number components. This error is caused by the superposition of high wave number components. If the high wave number components of the solution are sufficiently small compared to $N$ by keeping the cut-off wave number $N$ to be sufficiently large, it cannot be a significant problem. Therefore, pseudo-spectral method and collocation methods are sometimes preferred.

### 3. Discretization of Time Using Implicit Runge-Kutta Method

*3.1. Matrix Form*

As a preparation for the discretization of the time variables, Equation (7) is represented as a matrix form. Vectors $\mathbf{a}, \mathbf{b}, \mathbf{c}, \mathbf{d}$ are defined as follows.

$$
\begin{aligned}
\mathbf{a} &= (a_0, \ a_1, \ \cdots, \ a_N)^t, \\
\mathbf{b} &= (0, \ b_1, \ \cdots, \ b_N)^t, \\
\mathbf{c} &= (c_0, \ c_1, \ \cdots, \ c_N)^t, \\
\mathbf{d} &= (0, \ d_1, \ \cdots, \ d_N)^t.
\end{aligned}
\tag{8}
$$

Also, let us denote $g_0, h_0, g_l$ and $h_l$ $(l = 1, \cdots, N)$ by $g_0 = \frac{1}{J} \sum_{j=0}^{J-1} F(u_j)$, $h_0 = 0$, $g_l = \frac{2}{J} \sum_{j=0}^{J-1} F(u_j) \cos(\frac{2\pi l}{L} x_j)$ and $h_l = \frac{2}{J} \sum_{j=0}^{J-1} F(u_j) \sin(\frac{2\pi l}{L} x_j)$. We define $\mathbf{g}$ and $\mathbf{h}$ by

$$
\begin{aligned}
\mathbf{g} &= -\beta(g_0, \ g_1, \ \cdots, \ g_l, \ \cdots, \ g_N)^t, \\
\mathbf{h} &= -\beta(h_0, \ h_1, \ \cdots, \ h_l, \ \cdots, \ h_N)^t.
\end{aligned}
\tag{9}
$$

Furthermore, let us denote $\tilde{\alpha}_l = \alpha(\frac{2\pi l}{L})^2$ and $(N+1)$-order square matrix $A$, $E$, $E'$ by the following equation.

$$A = \begin{bmatrix} \tilde{\alpha}_0 & & & \\ & \tilde{\alpha}_1 & & \\ & & \ddots & \\ & & & \tilde{\alpha}_N \end{bmatrix}, \ E = \begin{bmatrix} 1 & & & \\ & 1 & & \\ & & \ddots & \\ & & & 1 \end{bmatrix}, \ E' = \begin{bmatrix} 0 & & & \\ & 1 & & \\ & & \ddots & \\ & & & 1 \end{bmatrix}, \quad (10)$$

Here, using Equations (8) to (10), the matrix representation of Equation (7) is as shown below.

$$\frac{d}{dt} \begin{bmatrix} \mathbf{a} \\ \mathbf{b} \\ \mathbf{c} \\ \mathbf{d} \end{bmatrix} = \begin{bmatrix} 0 & 0 & E & 0 \\ 0 & 0 & 0 & E' \\ A & 0 & 0 & 0 \\ 0 & A & 0 & 0 \end{bmatrix} \begin{bmatrix} \mathbf{a} \\ \mathbf{b} \\ \mathbf{c} \\ \mathbf{d} \end{bmatrix} + \begin{bmatrix} \mathbf{0} \\ \mathbf{0} \\ \mathbf{g} \\ \mathbf{h} \end{bmatrix}. \quad (11)$$

where $\mathbf{0} = (0, \ 0, \ \cdots, \ 0)^t$. Using the notations

$$\mathbf{W} = \begin{bmatrix} \mathbf{a} \\ \mathbf{b} \\ \mathbf{c} \\ \mathbf{d} \end{bmatrix}, \ \mathbf{F} = \begin{bmatrix} \mathbf{0} \\ \mathbf{0} \\ \mathbf{g} \\ \mathbf{h} \end{bmatrix}, \ \mathbf{M} = \begin{bmatrix} 0 & 0 & E & 0 \\ 0 & 0 & 0 & E' \\ A & 0 & 0 & 0 \\ 0 & A & 0 & 0 \end{bmatrix}, \quad (12)$$

Equation (11) is represented by

$$\frac{d}{dt}\mathbf{W} = \mathbf{MW} + \mathbf{F}(\mathbf{W}), \quad (13)$$

where $\mathbf{g}$, $\mathbf{h}$ depend on $\{u_i\}_{i=0}^{J-1}$, which is obtained by the inverse Fourier transform of $\mathbf{W}$. Therefore, after the spatial discretization, the inhomogeneous term generally holds the nonlinearity $\mathbf{F} = \mathbf{F}(\mathbf{W})$.

*3.2. Implicit Runge-Kutta Method*

Following the literature [18], we introduce the implicit Runge-Kutta method of two-stage and third-order for discretizing the time. Let $u(t)$ be the solution of the initial value problem of the abstract evolution equation.

$$\begin{cases} \frac{du}{dt} = f(t,u), & \alpha < t < \beta, \\ u(a) = u_0, \end{cases}$$

in a Hilbert space (corresponding to Equation (13)). The time interval $(\alpha, \beta)$ is divided into equally-discretized $M$ segments being incremented by $\Delta t = (\beta - \alpha)/M$. The discrete time sequence $\{t_m\}$ is represented by

$$t_m = \alpha + m\Delta t \ \ (m = 0, 1, \cdots, M),$$

and the approximated value of unknown function $u(t_m)$ is denoted by $U_m$. In this case, the method for obtaining the approximate value $U_{m+1}$ is called the Runge-Kutta method. More precisely, the Runge-Kutta method is represented by

$$\begin{cases} U_{m+1} = U_m + \Delta t \displaystyle\sum_{i=1}^{s} b_i l_i, \\ l_i = f(t_m + c_i \Delta t, \ U_m + \Delta t \sum_{j=1}^{s} a_{ij} l_j) \ \ (i = 1, 2, \cdots, s). \end{cases}$$

Here, the natural number $s$ is called the number of steps, and $a_{ij}$, $b_i$, $c_i$ are the parameters that define the formula. It is called the implicit Runge-Kutta method when $a_{ij} \neq 0$ $(j > i)$. The conditions

$$c_i = \sum_{j=1}^{s} a_{ij} \quad (i = 1, 2, \cdots, s)$$

are imposed on the parameters $a_{ij}$, $c_i$. The table of parameters $a_{ij}$, $b_i$, $c_i$

$$
\begin{array}{c|ccc}
c_1 & a_{11} & \cdots & a_{1s} \\
\vdots & \vdots & \ddots & \vdots \\
c_s & a_{s1} & \cdots & a_{ss} \\
\hline
& b_1 & \cdots & b_s
\end{array}
$$

is known as the Butcher tableau [18]. Under the assumption that $U_m = u(t_m)$, the local discretization error is defined by

$$T_{m+1} = \frac{1}{\Delta t} \left\{ u(t_{m+1}) - u(t_m) - \Delta t \sum_{i=1}^{s} b_i l_i \right\}.$$

Note that when the local discretization error is evaluates as $T_{n+1} = O((\Delta t)^p)$, the Runge-Kutta formula is said to be of order $p$. Although the number of stages and orders is arbitrary, in this paper, we adopt the Runge-Kutta method with two-stage and third-order by taking into account the balance of the accuracy and the calculation cost. The Butcher tableau for the implicit Runge-Kutta formula of two-stage and third-order is given by

$$
\begin{array}{c|cc}
1/3 & 5/12 & -1/12 \\
1 & 3/4 & 1/4 \\
\hline
& 3/4 & 1/4
\end{array}
$$

In general, explicit schemes have a restriction on the setting of the time spacing variables $\Delta t$, called the CFL condition. Therefore, it is important for numerical schemes to be A-stable. On the other hand, the Implicit Runge-Kutta Method is known to be A-stable. In this sense, the Implicit Runge-Kutta Method is preferably employed as a stable and high-order scheme [22–24]. However, the Implicit Runge-Kutta Method is not widely used. This is firstly due to the calculation cost; indeed the application of implicit schemes to nonlinear partial differential equations requires numerical iteration for each single time step. Another issue is that the convergence of the iterative method is not guaranteed when the degree of nonlinearity and the time increment range of the target problem are large. The numerical scheme proposed in this paper achieves a good balance between convergence (stability) and simplicity by using simple devices without applying complicated methods (cf. Section 3.4).

### 3.3. Application of Implicit Runge-Kutta Method and Iteration Formula

Applying the two-stage and third-order implicit Runge-Kutta method to discretize the time variables in (13), the resulting equations are shown by

$$
\begin{cases}
\mathbf{W}_{n+1} = \mathbf{W}_n + \frac{3}{4}\Delta t \mathbf{k}_1 + \frac{1}{4}\Delta t \mathbf{k}_2, \\
\mathbf{k}_1 = \mathbf{M}\left(\mathbf{W}_n + \frac{5}{12}\Delta t \mathbf{k}_1 - \frac{1}{12}\Delta t \mathbf{k}_2\right) + \mathbf{F}\left(\mathbf{W}_n + \frac{5}{12}\Delta t \mathbf{k}_1 - \frac{1}{12}\Delta t \mathbf{k}_2\right), \\
\mathbf{k}_2 = \mathbf{M}\left(\mathbf{W}_n + \frac{3}{4}\Delta t \mathbf{k}_1 + \frac{1}{4}\Delta t \mathbf{k}_2\right) + \mathbf{F}\left(\mathbf{W}_n + \frac{3}{4}\Delta t \mathbf{k}_1 + \frac{1}{4}\Delta t \mathbf{k}_2\right),
\end{cases}
\tag{14}
$$

where, using the $(N+1)$-dimensional vector $\mathbf{k}_i^a, \mathbf{k}_i^b, \mathbf{k}_i^c, \mathbf{k}_i^d$ $(i = 1, 2)$, vectors $\mathbf{k}_1$ and $\mathbf{k}_2$ are represented by

$$\mathbf{k}_1 = ((\mathbf{k}_1^a)^t, \ (\mathbf{k}_1^b)^t, \ (\mathbf{k}_1^c)^t, \ (\mathbf{k}_1^d)^t)^t,$$
$$\mathbf{k}_2 = ((\mathbf{k}_2^a)^t, \ (\mathbf{k}_2^b)^t, \ (\mathbf{k}_2^c)^t, \ (\mathbf{k}_2^d)^t)^t.$$

Time evolution of solution can be found by calculating $\mathbf{k}_1$ and $\mathbf{k}_2$, and substituting them into the first equation of (14). The second equation of (14) can be decomposed into four parts.

$$\begin{cases} \mathbf{k}_1^a = \mathbf{c} + \frac{5}{12}\Delta t \mathbf{k}_1^c - \frac{1}{12}\Delta t \mathbf{k}_2^c, \\ \mathbf{k}_1^b = E'\mathbf{d} + \frac{5}{12}\Delta t E' \mathbf{k}_1^d - \frac{1}{12}\Delta t E' \mathbf{k}_2^d, \\ \mathbf{k}_1^c = A\mathbf{a} + \frac{5}{12}\Delta t A \mathbf{k}_1^a - \frac{1}{12}\Delta t A \mathbf{k}_2^a + \mathbf{g}(\mathbf{W}_n + \frac{5}{12}\Delta t \mathbf{k}_1 - \frac{1}{12}\Delta t \mathbf{k}_2), \\ \mathbf{k}_1^d = A\mathbf{b} + \frac{5}{12}\Delta t A \mathbf{k}_1^b - \frac{1}{12}\Delta t A \mathbf{k}_2^b + \mathbf{h}(\mathbf{W}_n + \frac{5}{12}\Delta t \mathbf{k}_1 - \frac{1}{12}\Delta t \mathbf{k}_2). \end{cases} \tag{15}$$

Similarly, the third equation of (14) is decomposed into four parts.

$$\begin{cases} \mathbf{k}_2^a = \mathbf{c} + \frac{3}{4}\Delta t \mathbf{k}_1^c + \frac{1}{4}\Delta t \mathbf{k}_2^c, \\ \mathbf{k}_2^b = E'\mathbf{d} + \frac{3}{4}\Delta t E' \mathbf{k}_1^d + \frac{1}{4}\Delta t E' \mathbf{k}_2^d, \\ \mathbf{k}_2^c = A\mathbf{a} + \frac{3}{4}\Delta t A \mathbf{k}_1^a + \frac{1}{4}\Delta t A \mathbf{k}_2^a + \mathbf{g}(\mathbf{W}_n + \frac{3}{4}\Delta t \mathbf{k}_1 + \frac{1}{4}\Delta t \mathbf{k}_2), \\ \mathbf{k}_2^d = A\mathbf{b} + \frac{3}{4}\Delta t A \mathbf{k}_1^b + \frac{1}{4}\Delta t A \mathbf{k}_2^b + \mathbf{h}(\mathbf{W}_n + \frac{3}{4}\Delta t \mathbf{k}_1 + \frac{1}{4}\Delta t \mathbf{k}_2). \end{cases} \tag{16}$$

$\mathbf{k}_1, \mathbf{k}_2$ satisfying Equations (15) and (16) are found by means of an iterative method. Let the value of the $\nu$-th iteration be represented by

$$\mathbf{k}_1^\nu = ((\mathbf{k}_1^{a,\nu})^t, \ (\mathbf{k}_1^{b,\nu})^t, \ (\mathbf{k}_1^{c,\nu})^t, \ (\mathbf{k}_1^{d,\nu})^t)^t,$$
$$\mathbf{k}_2^\nu = ((\mathbf{k}_2^{a,\nu})^t, \ (\mathbf{k}_2^{b,\nu})^t, \ (\mathbf{k}_2^{c,\nu})^t, \ (\mathbf{k}_2^{d,\nu})^t)^t.$$

Then the formula for finding the $(\nu + 1)$-th value from the $\nu$-th, which corresponds to the transforms (15) and (16) are summarized as

$$\begin{cases} \mathbf{k}_1^{a,\nu+1} = \mathbf{c} + \frac{5}{12}\Delta t \mathbf{k}_1^{c,\nu} - \frac{1}{12}\Delta t \mathbf{k}_2^{c,\nu}, \\ \mathbf{k}_2^{a,\nu+1} = \mathbf{c} + \frac{3}{4}\Delta t \mathbf{k}_1^{c,\nu} + \frac{1}{4}\Delta t \mathbf{k}_2^{c,\nu}, \\ \mathbf{k}_1^{b,\nu+1} = E'\mathbf{d} + \frac{5}{12}\Delta t E' \mathbf{k}_1^{d,\nu} - \frac{1}{12}\Delta t E' \mathbf{k}_2^{d,\nu}, \\ \mathbf{k}_2^{b,\nu+1} = E'\mathbf{d} + \frac{3}{4}\Delta t E' \mathbf{k}_1^{d,\nu} + \frac{1}{4}\Delta t E' \mathbf{k}_2^{d,\nu}, \\ \mathbf{k}_1^{c,\nu+1} = A\mathbf{a} + \frac{5}{12}\Delta t A \mathbf{k}_1^{a,\nu} - \frac{1}{12}\Delta t A \mathbf{k}_2^{a,\nu} + \mathbf{g}(\mathbf{W}_n + \frac{5}{12}\Delta t \mathbf{k}_1^\nu - \frac{1}{12}\Delta t \mathbf{k}_2^\nu), \\ \mathbf{k}_2^{c,\nu+1} = A\mathbf{a} + \frac{3}{4}\Delta t A \mathbf{k}_1^{a,\nu} + \frac{1}{4}\Delta t A \mathbf{k}_2^{a,\nu} + \mathbf{g}(\mathbf{W}_n + \frac{3}{4}\Delta t \mathbf{k}_1^\nu + \frac{1}{4}\Delta t \mathbf{k}_2^\nu), \\ \mathbf{k}_1^{d,\nu+1} = A\mathbf{b} + \frac{5}{12}\Delta t A \mathbf{k}_1^{b,\nu} - \frac{1}{12}\Delta t A \mathbf{k}_2^{b,\nu} + \mathbf{h}(\mathbf{W}_n + \frac{5}{12}\Delta t \mathbf{k}_1^\nu - \frac{1}{12}\Delta t \mathbf{k}_2^\nu), \\ \mathbf{k}_2^{d,\nu+1} = A\mathbf{b} + \frac{3}{4}\Delta t A \mathbf{k}_1^{b,\nu} + \frac{1}{4}\Delta t A \mathbf{k}_2^{b,\nu} + \mathbf{h}(\mathbf{W}_n + \frac{3}{4}\Delta t \mathbf{k}_1^\nu + \frac{1}{4}\Delta t \mathbf{k}_2^\nu). \end{cases} \tag{17}$$

This is a fully discretized equation for both time and space.

### 3.4. Implementation of Iterative Method
3.4.1. Implementation of Half Step

To solve Equation (17), take $\mathbf{k}_1^1 = \mathbf{W}_n, \mathbf{k}_2^1 = \mathbf{W}_n$ as the initial value. Iteration is carried out until both $\mathbf{k}_1^\nu$ and $\mathbf{k}_2^\nu$ converge. In order to make the iteration process as short as possible

(i.e., $\mathbf{k}_1^\nu$ and $\mathbf{k}_2^\nu$ preferably converge in smaller iteration numbers). we introduce $\mathbf{k}_1^{\nu+1/2}$ and $\mathbf{k}_2^{\nu+1/2}$.

$$\mathbf{k}_1^{\nu+1/2} = ((\mathbf{k}_1^{a,\nu+1})^t, \ (\mathbf{k}_1^{b,\nu+1})^t, \ (\mathbf{k}_1^{c,\nu})^t, \ (\mathbf{k}_1^{d,\nu})^t)^t,$$
$$\mathbf{k}_2^{\nu+1/2} = ((\mathbf{k}_2^{a,\nu+1})^t, \ (\mathbf{k}_2^{b,\nu+1})^t, \ (\mathbf{k}_2^{c,\nu})^t, \ (\mathbf{k}_2^{d,\nu})^t)^t.$$

Accordingly Equation (17) is modified to

$$
\begin{cases}
\mathbf{k}_1^{a,\nu+1} = \mathbf{c} + \frac{5}{12}\Delta t \mathbf{k}_1^{c,\nu} - \frac{1}{12}\Delta t \mathbf{k}_2^{c,\nu}, \\[4pt]
\mathbf{k}_2^{a,\nu+1} = \mathbf{c} + \frac{3}{4}\Delta t \mathbf{k}_1^{c,\nu} + \frac{1}{4}\Delta t \mathbf{k}_2^{c,\nu}, \\[4pt]
\mathbf{k}_1^{b,\nu+1} = E'\mathbf{d} + \frac{5}{12}\Delta t E' \mathbf{k}_1^{d,\nu} - \frac{1}{12}\Delta t E' \mathbf{k}_2^{d,\nu}, \\[4pt]
\mathbf{k}_2^{b,\nu+1} = E'\mathbf{d} + \frac{3}{4}\Delta t E' \mathbf{k}_1^{d,\nu} + \frac{1}{4}\Delta t E' \mathbf{k}_2^{d,\nu}, \\[4pt]
\mathbf{k}_1^{c,\nu+1} = A\mathbf{a} + \frac{5}{12}\Delta t A\mathbf{k}_1^{a,\nu+1} - \frac{1}{12}\Delta t A\mathbf{k}_2^{a,\nu+1} \\[2pt]
\qquad\quad + \mathbf{g}(\mathbf{W}_n + \frac{5}{12}\Delta t \mathbf{k}_1^{\nu+1/2} - \frac{1}{12}\Delta t \mathbf{k}_2^{\nu+1/2}), \\[4pt]
\mathbf{k}_2^{c,\nu+1} = A\mathbf{a} + \frac{3}{4}\Delta t A\mathbf{k}_1^{a,\nu+1} + \frac{1}{4}\Delta t A\mathbf{k}_2^{a,\nu+1} \\[2pt]
\qquad\quad + \mathbf{g}(\mathbf{W}_n + \frac{3}{4}\Delta t \mathbf{k}_1^{\nu+1/2} + \frac{1}{4}\Delta t \mathbf{k}_2^{\nu+1/2}), \\[4pt]
\mathbf{k}_1^{d,\nu+1} = A\mathbf{b} + \frac{5}{12}\Delta t A\mathbf{k}_1^{b,\nu+1} - \frac{1}{12}\Delta t A\mathbf{k}_2^{b,\nu+1} \\[2pt]
\qquad\quad + \mathbf{h}(\mathbf{W}_n + \frac{5}{12}\Delta t \mathbf{k}_1^{\nu+1/2} - \frac{1}{12}\Delta t \mathbf{k}_2^{\nu+1/2}), \\[4pt]
\mathbf{k}_2^{d,\nu+1} = A\mathbf{b} + \frac{3}{4}\Delta t A\mathbf{k}_1^{b,\nu+1} + \frac{1}{4}\Delta t A\mathbf{k}_2^{b,\nu+1} \\[2pt]
\qquad\quad + \mathbf{h}(\mathbf{W}_n + \frac{3}{4}\Delta t \mathbf{k}_1^{\nu+1/2} + \frac{1}{4}\Delta t \mathbf{k}_2^{\nu+1/2}).
\end{cases}
\tag{18}
$$

In the following, we describe the procedure for calculating Equation (18) in detail. The calculation consists of two stages.

### 3.4.2. The First Stage

Let the $l$-th element of the vector $\mathbf{k}_x^{X,\nu}$ ($X = a, b, c, d;\ x = 1, 2$) be made of $k_{x,l}^{X,\nu}$ ($l = 0, 1, \cdots, N$), and let the $l$-th element of the vector $\mathbf{a}, \mathbf{b}, \mathbf{c}, \mathbf{d}$ be made of $a_l, b_l, c_l, d_l$ ($l = 0, 1, \cdots, N$) respectively. Calculations are performed for each components.

$$
\begin{cases}
k_{1,l}^{a,\nu+1} = c_l + \frac{5}{12}\Delta t k_{1,l}^{c,\nu} - \frac{1}{12}\Delta t k_{2,l}^{c,\nu}, \\[4pt]
k_{1,l}^{b,\nu+1} = d_l + \frac{5}{12}\Delta t k_{1,l}^{d,\nu} - \frac{1}{12}\Delta t k_{2,l}^{d,\nu}.
\end{cases}
\tag{19}
$$

$$
\begin{cases}
k_{2,l}^{a,\nu+1} = c_l + \frac{3}{4}\Delta t k_{1,l}^{c,\nu} + \frac{1}{4}\Delta t k_{2,l}^{c,\nu}, \\[4pt]
k_{2,l}^{b,\nu+1} = d_l + \frac{3}{4}\Delta t k_{1,l}^{d,\nu} + \frac{1}{4}\Delta t k_{2,l}^{d,\nu}.
\end{cases}
\tag{20}
$$

Using the results, we obtain $\mathbf{k}_1^{\nu+1/2}, \mathbf{k}_2^{\nu+1/2}$.

### 3.4.3. The Second Stage

Let the $l$-th element of the vector $\mathbf{g}, \mathbf{h}$ be represented by $\tilde{g}_l, \tilde{h}_l$ ($l = 0, 1, \cdots, N$), and calculate each component by the Fourier inverse transform using $\mathbf{k}_1^{\nu+1/2}, \mathbf{k}_2^{\nu+1/2}, \mathbf{W}_n$. Then, calculate the following equation using $\mathbf{k}_1^{\nu+1/2}, \mathbf{k}_2^{\nu+1/2}, \mathbf{W}_n, \tilde{g}_l, \tilde{h}_l$ ($l = 0, 1, \cdots, N$).

$$
\begin{cases}
k_{1,l}^{c,\nu+1} = \tilde{a}_l a_l + \frac{5}{12}\Delta t \tilde{a}_l k_{1,l}^{a,\nu+1} - \frac{1}{12}\Delta t \tilde{a}_l k_{2,l}^{a,\nu+1} \\
\qquad\quad +\tilde{g}_l(\mathbf{W}_n + \frac{5}{12}\Delta t \mathbf{k}_1^{\nu+1/2} - \frac{1}{12}\Delta t \mathbf{k}_2^{\nu+1/2}), \\
k_{1,l}^{d,\nu+1} = \tilde{a}_l b_l + \frac{5}{12}\Delta t \tilde{a}_l k_{1,l}^{b,\nu+1} - \frac{1}{12}\Delta t \tilde{a}_l k_{2,l}^{b,\nu+1} \\
\qquad\quad +\tilde{h}_l(\mathbf{W}_n + \frac{5}{12}\Delta t \mathbf{k}_1^{\nu+1/2} - \frac{1}{12}\Delta t \mathbf{k}_2^{\nu+1/2}).
\end{cases}
\tag{21}
$$

$$
\begin{cases}
k_{2,l}^{c,\nu+1} = \tilde{a}_l a_l + \frac{3}{4}\Delta t \tilde{a}_l k_{1,l}^{a,\nu+1} + \frac{1}{4}\Delta t \tilde{a}_l k_{2,l}^{a,\nu+1} \\
\qquad\quad +\tilde{g}_l(\mathbf{W}_n + \frac{3}{4}\Delta t \mathbf{k}_1^{\nu+1/2} + \frac{1}{4}\Delta t \mathbf{k}_2^{\nu+1/2}), \\
k_{2,l}^{d,\nu+1} = \tilde{a}_l b_l + \frac{3}{4}\Delta t \tilde{a}_l k_{1,l}^{b,\nu+1} + \frac{1}{4}\Delta t \tilde{a}_l k_{2,l}^{b,\nu+1} \\
\qquad\quad +\tilde{h}_l(\mathbf{W}_n + \frac{3}{4}\Delta t \mathbf{k}_1^{\nu+1/2} + \frac{1}{4}\Delta t \mathbf{k}_2^{\nu+1/2}).
\end{cases}
\tag{22}
$$

Using these results and $\mathbf{k}_1^{\nu+1/2}, \mathbf{k}_2^{\nu+1/2}$, we obtain $\mathbf{k}_1^{\nu+1}, \mathbf{k}_2^{\nu+1}$.

### 3.4.4. Decision of Convergence

If $\epsilon$ is a sufficiently small positive value (e.g., $10^{-12}$), and if the smaller value of the relative or absolute error between the $\nu$-th and $(\nu-1)$-th iterations is smaller than $\epsilon$, then the $\nu$-th iteration is regarded as converged. If $\mathbf{k}_1^{\nu+1}$ and $\mathbf{k}_2^{\nu+1}$ are not converged, return to the first stage of calculation. If $\mathbf{k}_1^{\nu+1}, \mathbf{k}_2^{\nu+1}$ is convergent, then $\mathbf{W}_{n+1}$ is obtained using the formula

$$
\mathbf{W}_{n+1} = \mathbf{W}_n + \frac{3}{4}\Delta t \mathbf{k}_1^{\nu+1} + \frac{1}{4}\Delta t \mathbf{k}_2^{\nu+1}.
\tag{23}
$$

The calculation cost of this numerical scheme applying the implicit Runge-Kutta method and the spectral method is estimated to be $O(N\log_2 N)$, and the local discretization error of the time variable is estimated to be of third-order.

## 4. Benchmark Calculations

### 4.1. Linear Case

#### 4.1.1. Comparison to Exact Solution

Let us take the initial and boundary value problem (2) with $F(u) = u, \alpha = -1, \beta = 1$, $\Omega = [0, L]$.

$$
\begin{cases}
\frac{\partial v}{\partial t} - \frac{\partial^2 u}{\partial x^2} + u = 0, \\
\frac{\partial u}{\partial t} = v, \\
u(x,0) = 0, \ u(0,t) = u(L,t), \\
v(x,0) = \cos(\frac{2\pi}{L}x), \ v(0,t) = v(L,t).
\end{cases}
\tag{24}
$$

The exact solution to this problem is given by

$$
\begin{cases}
u(x,t) = \dfrac{1}{\sqrt{1+\left(\frac{2\pi}{L}\right)^2}} \sin\left(\sqrt{1+\left(\frac{2\pi}{L}\right)^2}\, t\right) \cos\left(\frac{2\pi}{L}x\right), \\
v(x,t) = \cos\left(\sqrt{1+\left(\frac{2\pi}{L}\right)^2}\, t\right) \cos\left(\frac{2\pi}{L}x\right).
\end{cases}
\tag{25}
$$

For Equation (24), the time evolutions of the numerical solution $u, v$ ($J \geq 2N + 1, L = 8$) until $t = 1$ are shown in Figure 1, and those at $t = 1$ are shown in Figure 2. Since the exact solution is represented by Equation (25), it is possible to evaluate the accuracy depending on the spatio-temporal discretization. Furthermore, in terms of clarifying the merit of the proposed scheme, the error between the numerical results by the $\theta$ method with $\theta = 1/2$ (for the preceding calculation using the $\theta$ method in time, see [21]) are also calculated.

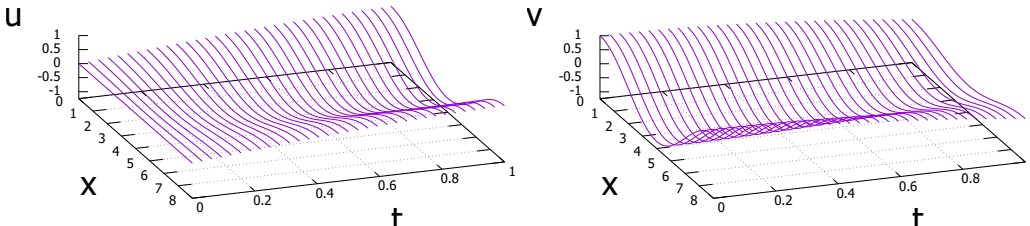

**Figure 1.** Linear Klein-Gordon dynamics: time evolution of $u$ and $v$ ($N = 2^{10}$, $\Delta t = 2^{-13}$, $L = 8$).

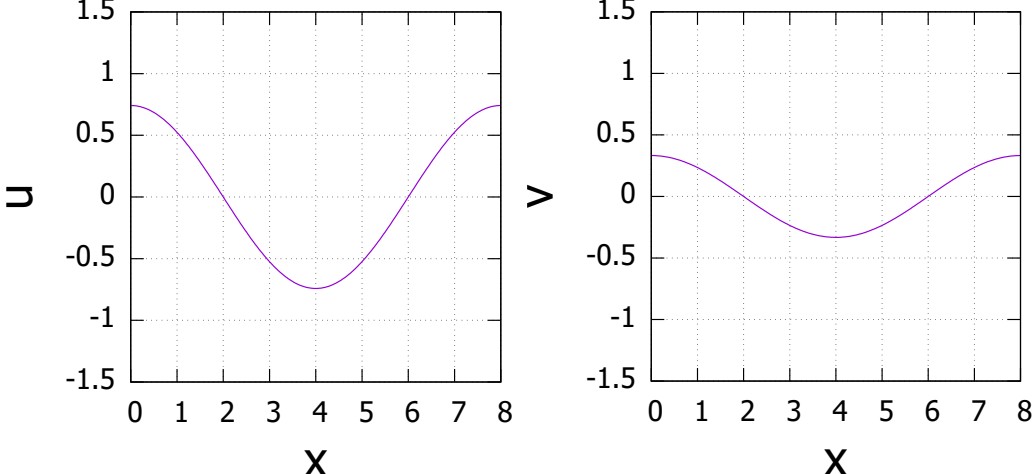

**Figure 2.** Solution of linear Klein-Gordon equation at $t = 1$ ($N = 2^{10}$, $\Delta t = 2^{-13}$, $L = 8$).

The comparison between the numerical and the exact solutions are shown in Figures 3 and 4. In the present paper, the error is defined as the smaller one of the relative and absolute errors. In Figure 3, numerical solutions with different time spacing unit $\Delta t$ are examined. The maximum error, which is determined by running for $x_j$ ($j = 0, \cdots, J$), between the numerical and the exact solutions are calculated. We see that exponential dependence on the time discretization is noticed. It also shows that the high accuracy of implicit Runge-Kutta method; indeed at $\Delta t < 10^{-4}$, almost $10^{-4}$ times accurate result can be obtained compared to the $\theta$-method with $\theta = 1/2$ (the Crank-Nicolson method). In Figure 4, numerical solutions with different spatial spacing parameter $N$ are examined. The maximum error, which is determined by running for $x_j$ ($j = 0, \cdots, J$), between the numerical and the exact solutions are calculated. We see that in the logarithmic scale, there is no significant $N$ dependence in calculations of any kinds.

4.1.2. Accuracy Depending on Discretization of Time Variables

Figure 3 shows $\Delta t$-dependence of errors. As for the two-stage and third-order implicit Runge-Kutta method ($+$ and $\circ$ in the figure), we see that the error becomes $1/8$ times smaller if $\Delta t$ is taken to be $1/2$ times smaller. In other words, the numerical results confirm that the scheme holds the third-order accuracy with respect to time. On the other hand, as for the $\theta$ method ($\times$ and $\square$ in the figure), we see that the error becomes $1/4$ times smaller if $\Delta t$ is taken to be $1/2$ times smaller.

For each of the above two numerical schemes, by comparing the errors in case of $N = 2^5$ and $N = 2^{10}$, the values of the errors are almost unchanged if the amplitude of $\Delta t$ is the same (note that in case of $N = 2^{10}$, the numerical calculation does not converge and no numerical solution is obtained for large $\Delta t$). That is, if $N$ is taken to be sufficiently large, the error depends only on the size of $\Delta t$ and not on the size of $N$. If we limit ourselves to the linear case of our benchmark calculations, this advantage arises essentially from the introduction of the Fourier spectral method for the spatial direction.

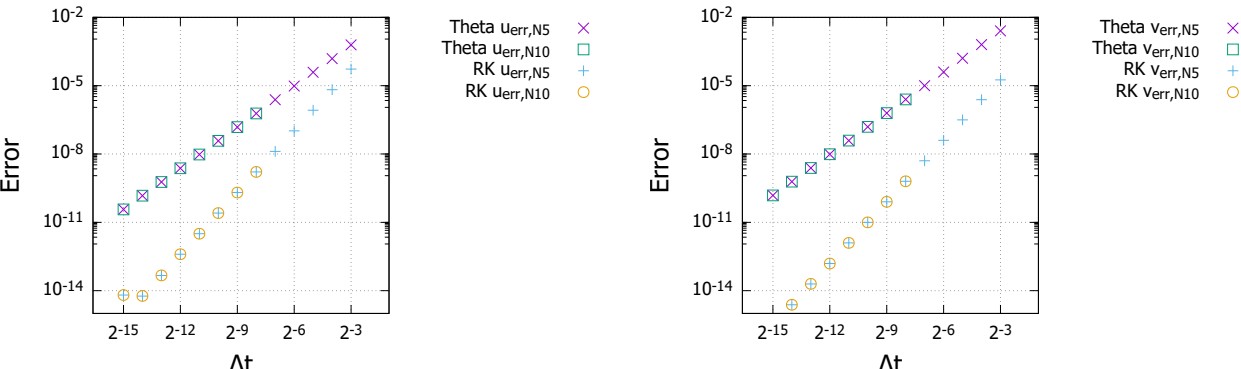

**Figure 3.** The relation between the error and the time increments $\Delta t$ for $u$ and $v$ at $t = 1$. Theta $u_{err,N5}$, Theta $u_{err,N10}$, Theta $v_{err,N5}$ and Theta $v_{err,N10}$ in the figure show the error of $u$ and $v$, when the $\theta$ method with $\theta = 1/2$ is applied with spatial spacing parameter $N = 2^5$ and $2^{10}$, respectively. Similarly, RK $u_{err,N5}$ and RK $u_{err,N10}$, RK $v_{err,N5}$ and RK $v_{err,N10}$ denote the error of $u$ and $v$, when the implicit Runge-Kutta method is applied with $N = 2^5$ and $2^{10}$, respectively.

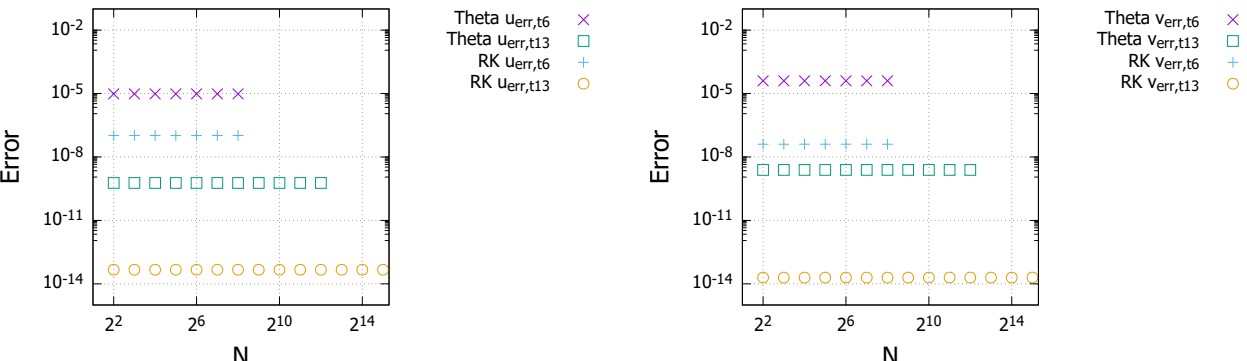

**Figure 4.** The relation between the error and the spatial spacing parameter $N$ for $u$ and $v$ at $t = 1$. Theta $u_{err,t6}$, Theta $u_{err,t13}$, Theta $v_{err,t6}$, and Theta $v_{err,t13}$ in the figure show the error in $u$ and $v$, when the $\theta$ method is applied with the time increments $\Delta t = 2^{-6}, 2^{-13}$, respectively. Similarly, RK $u_{err,t6}$, RK $u_{err,t16}$, RK $v_{err,t6}$ and RK $v_{err,t16}$ denote the error in $u$ and $v$, when the implicit Runge-Kutta method is applied with the time increments $\Delta t = 2^{-6}, 2^{-13}$, respectively.

In any case, the error depends only on $\Delta t$ for sufficiently large $N$, and the smaller $\Delta t$ results in the smaller error. Comparing two different schemes, it is concluded in the linear case of benchmark that the implicit Runge-Kutta method possibly includes $10^{-4}$ times smaller errors compared to the $\theta$ method.

### 4.1.3. Accuracy Depending on Discretization of Spatial Variables

Figure 4 shows that the error does not depend on $N = 2^2, 2^3, \cdots, 2^{15}$ (refer to $+$ and $\circ$ in Figure 4 for the errors by implicit Runge-Kutta method, and to $\times$ and $\square$ for those by $\theta$ method). Regardless of the choice of $N$, it is confirmed in Figure 4 that the errors with $\Delta t = 2^{-13}$ are smaller than those with $\Delta t = 2^{-6}$ (note that in case of $\Delta t = 2^{-6}$, the numerical calculation does not converge and no numerical solution is obtained for large $N$). This means that the error depends only on $\Delta t$ and not on $N$.

In conclusion, most of the error arises from the discretization of time. Therefore, if we limit ourselves to the benchmark (the linear case), for example, by taking $\Delta t \sim 10^{-4}$ and $N \sim 2^5$, we can obtain high-precision numerical solutions.

4.1.4. Convergence/Stability of Iteration

The numerical scheme proposed in this paper is A-stable, because it employs the implicit Runge-Kutta method. However, the problem to be solved by the implicit scheme results in numerical iterations. Since the convergence of the iterative method is not guaranteed in general, the convergence of the iterative method and the speed of convergence, which indicate a kind of stability in the case of applying implicit Runge-Kutta methods, must be discussed. Here, for improving convergence and accelerating an iterative process, the intermediate step shown in Equation (18) is equipped. This iterative method is referred to as the modified iterative method in this paper. In order to examine the efficiency of the modified iterative method, we compare the average number of iterations in calculating the time evolution of Equation (24) up to $t = 1$ with several $N$ and $\Delta t$. In addition, the results of applying the iterative method with Equation (17) (normal iterative method) are also shown for comparison.

First, Table 1 shows the average number of iterations when $N = 2^{10}$ is fixed and $\Delta t = 2^{-2}, 2^{-3}, \cdots, 2^{-10}$. All the calculations by the modified iterative method converge in 4 iterations when $\Delta t = 2^{-8}, 2^{-9}, 2^{-10}$. On the contrary, the corresponding results by the normal iterative method have 6 or 5 iterations, respectively. When $\Delta t = 2^{-2}, 2^{-3}, \cdots, 2^{-7}$, both modified iterative method and normal iterative method do not converge. Next, Table 2 shows the average number of iterations when $N = 2^5$ is fixed and $\Delta t = 2^{-2}, 2^{-3}, \cdots, 2^{-10}$. For $\Delta t = 2^{-3}$, the modified iterative method converges in 7 iterations, and the smaller $\Delta t$ leads to the fewer convergent iterations. The same trend can be seen also in the case of the normal iterative method. However, when $\Delta t = 2^{-2}$, both modified iterative method and normal iterative method do not converge. Accordingly, by applying the same convergence condition

$$\frac{\Delta t}{N^{-1}} \leq 2^2, \tag{26}$$

it is confirmed that the modified iterative method reduces the number of iterations by 20–30%, and the calculation cost is reduced significantly. Here not that the condition (26) plays a role of the CFL condition in the implicit Runge-Kutta cases.

**Table 1.** An average number of iterations by the iterative method with fixed $N = 2^{10}$. Modified ite. count and Normal ite. count denotes the average number of iterations by the modified iterative method and the normal iterative method, respectively. N/A denotes the case where the iterations did not converge.

| $N$ | $\Delta t$ | $\Delta t/N^{-1}$ | (a) Modified Ite. Count | (b) Normal Ite. Count | (a)/(b) |
|---|---|---|---|---|---|
| $2^{10}$ | $2^{-2}$ | $2^8$ | N/A | N/A | N/A |
| $2^{10}$ | $2^{-3}$ | $2^7$ | N/A | N/A | N/A |
| $2^{10}$ | $2^{-4}$ | $2^6$ | N/A | N/A | N/A |
| $2^{10}$ | $2^{-5}$ | $2^5$ | N/A | N/A | N/A |
| $2^{10}$ | $2^{-6}$ | $2^4$ | N/A | N/A | N/A |
| $2^{10}$ | $2^{-7}$ | $2^3$ | N/A | N/A | N/A |
| $2^{10}$ | $2^{-8}$ | $2^2$ | 4 | 6 | 0.67 |
| $2^{10}$ | $2^{-9}$ | $2^1$ | 4 | 6 | 0.67 |
| $2^{10}$ | $2^{-10}$ | $2^0$ | 4 | 5 | 0.80 |

**Table 2.** Average of the number of iterations by the iterative method with fixed $N = 2^5$. Modified ite. count and Normal ite. count denotes the average number of iterations by the modified iterative method and the normal iterative method, respectively. N/A denotes the case where the iterations did not converge.

| $N$ | $\Delta t$ | $\Delta t / N^{-1}$ | (a) Modified Ite. Count | (b) Normal Ite. Count | (a)/(b) |
|-----|-----|-----|-----|-----|-----|
| $2^5$ | $2^{-2}$ | $2^3$ | N/A | N/A | N/A |
| $2^5$ | $2^{-3}$ | $2^2$ | 7 | 11 | 0.64 |
| $2^5$ | $2^{-4}$ | $2^1$ | 6 | 9 | 0.67 |
| $2^5$ | $2^{-5}$ | $2^0$ | 5 | 8 | 0.63 |
| $2^5$ | $2^{-6}$ | $2^{-1}$ | 5 | 7 | 0.71 |
| $2^5$ | $2^{-7}$ | $2^{-2}$ | 4 | 7 | 0.57 |
| $2^5$ | $2^{-8}$ | $2^{-3}$ | 4 | 6 | 0.67 |
| $2^5$ | $2^{-9}$ | $2^{-4}$ | 4 | 6 | 0.67 |
| $2^5$ | $2^{-10}$ | $2^{-5}$ | 4 | 5 | 0.80 |

*4.2. Nonlinear Case*

4.2.1. Comparison to Exact Solution

Let us take the initial and boundary value problem (2) with $F(u) = \sin u$, $\alpha = -1$, $\beta = 1$, $\Omega = [0, L]$. The nonlinear wave equation in this case is known as the Sine-Gordon equation.

$$
\begin{cases}
\frac{\partial v}{\partial t} - \frac{\partial^2 u}{\partial x^2} + \sin u = 0, \\[2mm]
\frac{\partial u}{\partial t} = v, \\[2mm]
v(x,0) = -\sqrt{2}\, \frac{\operatorname{cn}(x,\frac{1}{2})\,\operatorname{dn}(x,\frac{1}{2})}{\sqrt{1-\frac{1}{4}\operatorname{sn}(x,\frac{1}{2})}}, \quad v(0,t) = v(L,t), \\[3mm]
u(x,0) = 2\sin^{-1}\left[\frac{1}{2}\operatorname{sn}(x,\frac{1}{2})\right], \quad u(0,t) = u(L,t).
\end{cases}
\tag{27}
$$

The exact solution to this problem is given by

$$
\begin{cases}
u(x,t) = \quad 2\sin^{-1}\left[\frac{1}{2}\operatorname{sn}(x - \sqrt{2}t, \frac{1}{2})\right], \\[3mm]
v(x,t) = \quad -\sqrt{2}\, \frac{\operatorname{cn}(x-\sqrt{2}t,\frac{1}{2})\,\operatorname{dn}(x-\sqrt{2}t,\frac{1}{2})}{\sqrt{1-\frac{1}{4}\operatorname{sn}^2(x-\sqrt{2}t,\frac{1}{2})}},
\end{cases}
\tag{28}
$$

where $\operatorname{sn}, \operatorname{cn}, \operatorname{dn}$ are Jacobi's elliptic functions, and $L$ can be expressed using complete elliptic integral of the first kind (See Appendix A and textbook [25]).

$$
\begin{cases}
L = 4F\left(\frac{\pi}{2}, \frac{1}{2}\right) = 4\int_0^{\pi/2} \frac{1}{\sqrt{1-\left(\frac{1}{2}\right)^2 \sin^2\theta}}\, d\theta \\[4mm]
= 6.743001419250385098\cdots
\end{cases}
\tag{29}
$$

For Equation (27), the time evolutions of the solution $u$, $v$ (with $J \geq 2N + 1$, $L = 6.7430014192503$) until $t = 1$ are shown in Figure 5, and those at time $t = 1$ are shown in Figure 6. Since the exact solution is represented by Equation (28), it is possible to evaluate the accuracy depending on the spatio-temporal discretization. Furthermore, similar to the linear case, the error between the numerical results by the $\theta$ method with $\theta = 1/2$ are also calculated.

The comparison between the numerical and the exact solutions are shown in Figures 7 and 8. In Figure 7, numerical solutions with different time spacing unit $\Delta t$ are examined. The maximum error, which is determined by running for $x_j$ $(j = 0, \cdots, J)$, between the numerical and the exact solutions are calculated. The exponential dependence on the time discretization is also noticed in the nonlinear case. It also shows that the high accuracy of implicit Runge-Kutta method; indeed at $\Delta t < 10^{-4}$, almost $10^{-4}$ times accurate result can be obtained compared to the $\theta$-method with $\theta = 1/2$. In Figure 8, numerical solutions with different spatial spacing parameters $N$ are examined. The maximum error, which is determined by running for $x_j$ $(j = 0, \cdots, J)$, between the numerical and the exact solutions are calculated. We see that, in the logarithmic scale, there is no significant $N$ dependence when $N$ is sufficiently large.

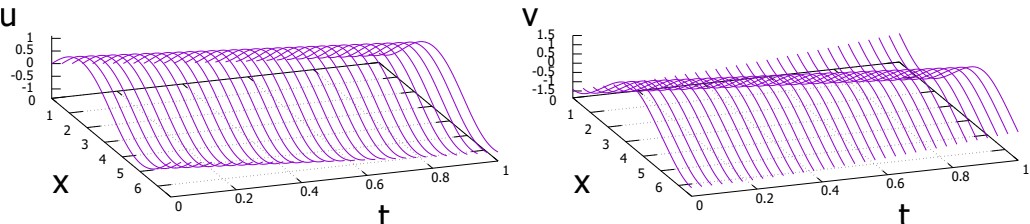

**Figure 5.** Nonlinear Klein-Gordon dynamics: time evolution of $u$ and $v$ ($N = 2^{10}$, $\Delta t = 2^{-13}$, $L = 6.7430014192503$).

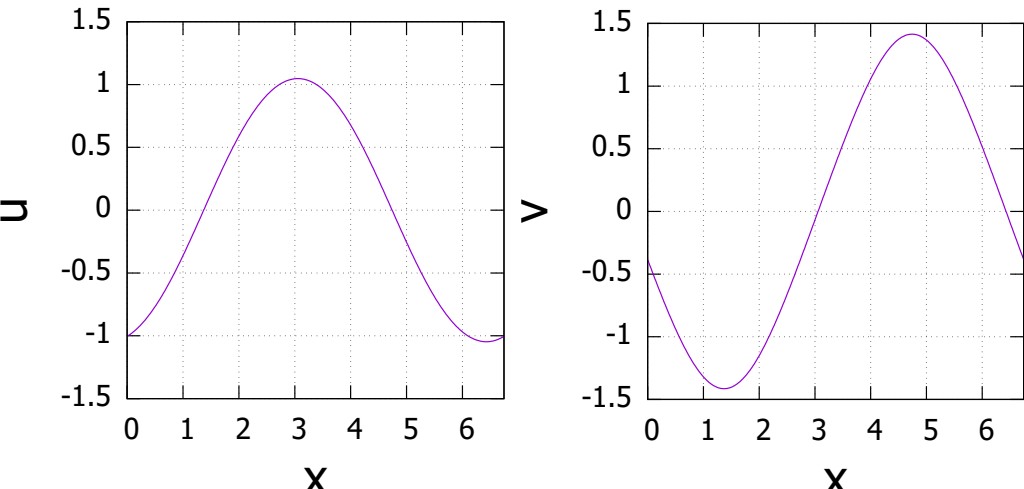

**Figure 6.** Solution of nonlinear Klein-Gordon equation: $u$ and $v$ at $t = 1$ ($N = 2^{10}$, $\Delta t = 2^{-13}$, $L = 6.7430014192503$).

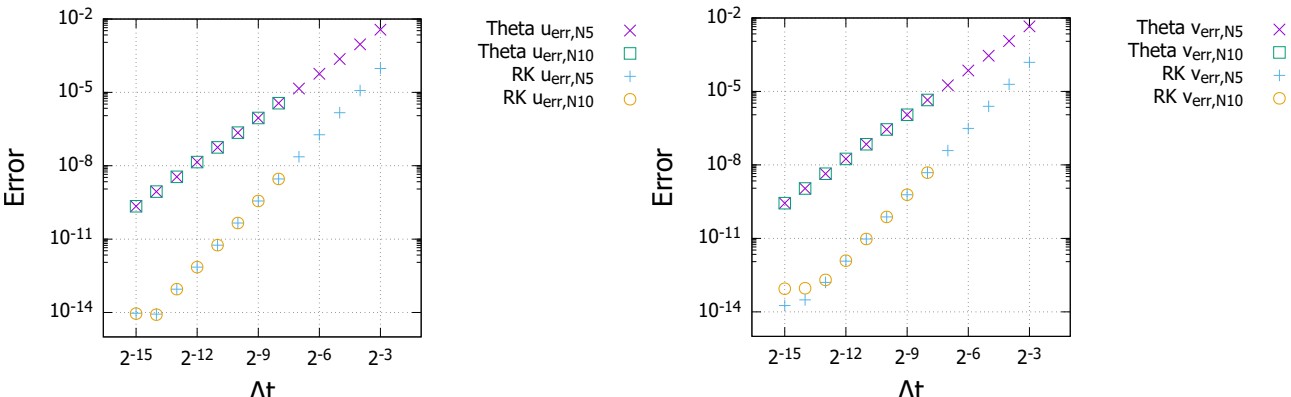

**Figure 7.** The relation between the error and the time increments $\Delta t$ for $u$ and $v$ at $t = 1$. Theta $u_{err,N5}$, Theta $u_{err,N10}$, Theta $v_{err,N5}$ and Theta $v_{err,N10}$ in the figure show the error of $u$ and $v$, when the $\theta$ method with $\theta = 1/2$ is applied with spatial spacing parameter $N = 2^5$ and $2^{10}$, respectively. Similarly, RK $u_{err,N5}$ and RK $u_{err,N10}$, RK $v_{err,N5}$ and RK $v_{err,N10}$ denote the error of $u$ and $v$, when the implicit Runge-Kutta method is applied with $N = 2^5$ and $2^{10}$, respectively.

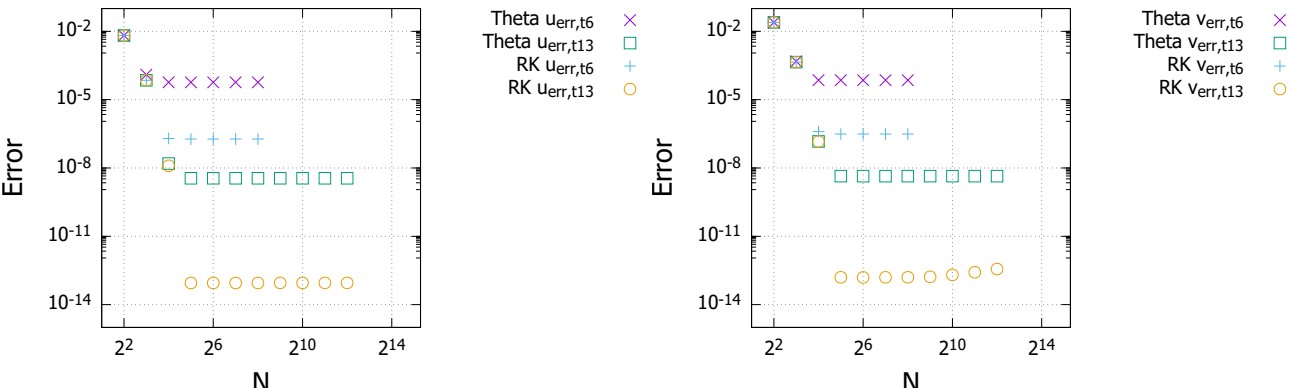

**Figure 8.** The relation between the error and the spatial spacing parameter $N$ for $u$ and $v$ at $t = 1$. Theta $u_{err,t6}$, Theta $u_{err,t13}$, Theta $v_{err,t6}$, and Theta $v_{err,t13}$ in the figure show the error in $u$ and $v$, when the $\theta$ method is applied with the time increments $\Delta t = 2^{-6}, 2^{-13}$, respectively. Similarly, RK $u_{err,t6}$, RK $u_{err,t16}$, RK $v_{err,t6}$ and RK $v_{err,t16}$ denote the error in $u$ and $v$, when the implicit Runge-Kutta method is applied with the time increments $\Delta t = 2^{-6}, 2^{-13}$, respectively.

### 4.2.2. Accuracy Depending on Discretization of Time Variables

Figure 7 shows $\Delta t$-dependence of errors. As for the two-stage and third-order implicit Runge-Kutta method ($+$ and $\circ$ in the figure), we see that the error becomes $1/8$ times smaller if $\Delta t$ is taken to be $1/2$ times smaller. In other words, even in the nonlinear case, the numerical results confirm that the scheme holds the third-order accuracy with respect to time. Also in nonlinear cases, as for the $\theta$ method ($\times$ and $\square$ in the figure), we see that the error becomes $1/4$ times smaller if $\Delta t$ is taken to be $1/2$ times smaller.

For each of the above two numerical schemes, by comparing the errors in case of $N = 2^5$ and $N = 2^{10}$, the values of the errors are almost unchanged if the amplitude of $\Delta t$ is the same (note that in case of $N = 2^{10}$, the numerical calculation does not converge and no numerical solution is obtained for large $\Delta t$). That is, if $N$ is taken to be sufficiently large, the error depends only on the size of $\Delta t$ and not on the size of $N$. If we limit ourselves to our benchmark calculations(also in the nonlinear case), this advantage arises essentially from the introduction of the Fourier spectral method for the spatial direction.

In any case, the error depends only on $\Delta t$ for sufficiently large $N$, and the smaller $\Delta t$ results in the smaller error. Comparing two different schemes, it is concluded also in the nonlinear case of benchmark that the implicit Runge-Kutta method possibly includes almost $10^{-4}$ times smaller errors compared to the $\theta$ method.

### 4.2.3. Accuracy Depending on Discretization of Spatial Variables

Figure 8 shows that if $N$ is larger than $2^5$, the error does not depend on the size of $N$ for either $\Delta t = 2^{-6}, 2^{-13}$ (refer to $+, \circ$ in Figure 8 for the errors by implicit Runge-Kutta method, and to $\times, \square$ for those by $\theta$ method). In addition, the smaller $\Delta t$ leads to a smaller error. On the other hand, if $N$ is smaller than $2^3$, the smaller $N$ leads to the larger error in which the order of error is almost the same regardless of the choice of the scheme and the amplitude of $\Delta t$. That is, most of the error arises from the discretization of space variables if $N$ is smaller than $2^3$, while most of the error arises from the discretization of time variables if $N$ is larger than $2^5$.

In conclusion, if $N$ is sufficiently large, the error associated with the discretization of spatial variables becomes sufficiently small, and most of the error arises from the discretization of time. Therefore, similar to the linear case, the benchmark calculations suggest that high-precision numerical solutions can be obtained by taking $\Delta t \sim 10^{-4}$ and $N \sim 2^5$.

### 4.2.4. Convergence/Stability of Iteration

As in the linear case, in order to confirm the properties of the iterative method from the viewpoint of stability in a broad sense, we present the results of the average number of iterations in calculating the time evolution of Equation (27) up to time $t = 1$ with several $N$ and $\Delta t$. In addition, the results of applying the iterative method with Equation (17) (normal iterative method) are also shown for comparison.

First, Table 3 shows the average number of iterations when $N = 2^{10}$ is fixed and $\Delta t = 2^{-2}, 2^{-3}, \cdots, 2^{-10}$. For $\Delta t = 2^{-8}, 2^{-9}, 2^{-10}$, the calculations by the modified iterative method converge in 11, 4, 4 iterations, respectively. On the contrary, the corresponding results by the normal iterative method have 16, 6, and 5 iterations, respectively. When $\Delta t = 2^{-2}, 2^{-3}, \cdots, 2^{-7}$, both modified iterative method and normal iterative method do not converge. Next, Table 4 shows the average number of iterations when $N = 2^5$ and $\Delta t = 2^{-2}, 2^{-3}, \cdots, 2^{-10}$. For $\Delta t = 2^{-3}$, the modified iterative method converges in 13 iterations, and the smaller $\Delta t$ leads to the fewer convergent iterations. The same trend can be seen also in the case of the normal iterative method. However, when $\Delta t = 2^{-2}$, both modified iterative method and normal iterative method do not converge.

Accordingly, by applying the same convergence condition

$$\frac{\Delta t}{N^{-1}} \leq 2^2$$

it is confirmed that the modified iterative method reduces the number of iterations by 20–50%, and the calculation cost is reduced significantly.

**Table 3.** Average number of iterations by the iterative method with fixed $N = 2^{10}$. Modified ite. count and Normal ite. count denotes the average number of iterations by the modified iterative method and the normal iterative method, respectively. N/A denotes the case where the iterations did not converge.

| $N$ | $\Delta t$ | $\Delta t / N^{-1}$ | (a) Modified Ite. Count | (b) Normal Ite. Count | (a)/(b) |
|---|---|---|---|---|---|
| $2^{10}$ | $2^{-2}$ | $2^8$ | N/A | N/A | N/A |
| $2^{10}$ | $2^{-3}$ | $2^7$ | N/A | N/A | N/A |
| $2^{10}$ | $2^{-4}$ | $2^6$ | N/A | N/A | N/A |
| $2^{10}$ | $2^{-5}$ | $2^5$ | N/A | N/A | N/A |
| $2^{10}$ | $2^{-6}$ | $2^4$ | N/A | N/A | N/A |
| $2^{10}$ | $2^{-7}$ | $2^3$ | N/A | N/A | N/A |
| $2^{10}$ | $2^{-8}$ | $2^2$ | 11 | 16 | 0.69 |
| $2^{10}$ | $2^{-9}$ | $2^1$ | 4 | 6 | 0.67 |
| $2^{10}$ | $2^{-10}$ | $2^0$ | 4 | 5 | 0.80 |

**Table 4.** Average of the number of iterations by the iterative method with fixed $N = 2^5$. Modified ite. count and Normal ite. count denotes the average number of iterations by the modified iterative method and the normal iterative method, respectively. N/A denotes the case where the iterations did not converge.

| $N$ | $\Delta t$ | $\Delta t/N^{-1}$ | (a) Modified Ite. Count | (b) Normal Ite. Count | (a)/(b) |
|---|---|---|---|---|---|
| $2^5$ | $2^{-2}$ | $2^3$ | N/A | N/A | N/A |
| $2^5$ | $2^{-3}$ | $2^2$ | 13 | 24 | 0.54 |
| $2^5$ | $2^{-4}$ | $2^1$ | 8 | 14 | 0.57 |
| $2^5$ | $2^{-5}$ | $2^0$ | 6 | 10 | 0.60 |
| $2^5$ | $2^{-6}$ | $2^{-1}$ | 6 | 9 | 0.67 |
| $2^5$ | $2^{-7}$ | $2^{-2}$ | 5 | 7 | 0.71 |
| $2^5$ | $2^{-8}$ | $2^{-3}$ | 4 | 7 | 0.57 |
| $2^5$ | $2^{-9}$ | $2^{-4}$ | 4 | 6 | 0.67 |
| $2^5$ | $2^{-10}$ | $2^{-5}$ | 4 | 5 | 0.80 |

## 5. Summary

In order to understand the dynamics of nonlinear hyperbolic equations, the numerical approach is one of the most efficient tools, where the smoothing effect is hopeless for generic hyperbolic equations. In this sense, accuracy is an indispensable factor in any way (for the conservation of physical quantities of the present scheme, see [26,27]). A high precision numerical scheme for nonlinear hyperbolic evolution equations is proposed, and its performance is examined by linear and nonlinear benchmark calculations. The numerical scheme consists of the implicit Runge-Kutta method and the Fourier spectral method. For a concrete example, the demonstration of scheme is carried out for one-dimensional Klein-Gordon equations. The precision due to the implicit Runge-Kutta method and spectral method is quantitatively shown by the errors of benchmark calculations in comparison to the $\theta$ method, and in comparison to the difference of time spacing variables.

As demonstrated in benchmark cases, it is confirmed that the scheme constructed in this paper has a third-order accuracy with respect to time. We have also confirmed that the truncation error associated with the spatial discretization is much smaller than the error associated with the time discretization by setting $N$ to be sufficiently large. That is, for sufficiently large $N$, most of the error arises from the error associated with $\Delta t$.

In time discretization, due to the limitation of the time spacing variables by the CFL condition, it is generally difficult to achieve both small error and low computational cost simply by the conventional explicit method. Furthermore, in spatial discretization, due to the effects of numerical dispersion, it is generally difficult for the typical finite difference method to preserve the conserved quantities to a sufficient degree. The proposed method settles these two points by simultaneously introducing the implicit Runge-Kutta method and the spectral method. The applicability of the scheme is confirmed by its computational order $O(N \log_2 N)$. In conclusion, the proposed scheme provides a generic framework for high-precision computation with relatively low computational cost.

From a future perspective, more complex problems with various boundary conditions and the behavior of solutions with discontinuities, construction of numerical scheme employing the spectral element method and/or the spectral penalty method is a promising direction. The proposed scheme should be a steadfast basis for such future works.

**Author Contributions:** Design of the study is prepared by all the authors. The first author (Y.T.) constructed the scheme and performed numerical experiments. The second author (Y.I.) drafted the manuscript. Y.T. and Y.I. All authors have read and agreed to the published version of the manuscript.

**Funding:** The authors have received no funding for this work.

**Institutional Review Board Statement:** Not applicable.

**Informed Consent Statement:** Not applicable.

**Data Availability Statement:** Not applicable.

**Conflicts of Interest:** The authors declare that there is no conflict of interest.

## Appendix A. Jacobian Elliptic Functions

A set of basic elliptic functions was introduced by Carl Gustav Jacob Jacobi [28] in 1829. These functions are named the Jacobian elliptic functions after him. For any $k \in [0,1)$, we define $K(k)$ by the incomplete elliptic integral of the first kind.

$$K(k) = \int_0^1 \frac{dt}{\sqrt{(1-t^2)(1-k^2t^2)}}. \tag{A1}$$

Then, for any $k \in [0,1)$ and any $x \in [-K(k), K(k)]$, we define $\text{sn}(x,k)$ by an inverse of the incomplete elliptic integral of the first kind.

$$x = \int_0^{sn(x,k)} \frac{dt}{\sqrt{(1-t^2)(1-k^2t^2)}}. \tag{A2}$$

Clearly, $\text{sn}(x,k)$ is an increasing odd function in $x$ from $[-K(k), K(k)]$ to $[-1,1]$. We extend the domain of $\text{sn}(x,k)$ to $\mathbb{R}$ by $\text{sn}(x+2K(k),k) = -\text{sn}(x,k)$, which implies that $\text{sn}(x,k)$ has $4K(k)$-periodicity. We can see that $K(0) = \pi/2$, $\text{sn}(x,0) = \sin x$, $K(k) \to \infty$, $\text{sn}(x,k) \to \tanh x$ as $k \to 1$ and $\text{sn}(\cdot,k) \in C^\infty(\mathbb{R})$. Using $\text{sn}(x,k)$, for $x \in [-K(k), K(k)]$, we also define

$$\text{cn}(x,k) = \sqrt{1 - \text{sn}^2(x,k)}, \tag{A3}$$

$$\text{dn}(x,k) = \sqrt{1 - k^2\text{sn}^2(x,k)}. \tag{A4}$$

Clearly, $\text{cn}(x,k)$ and $\text{dn}(x,k)$ are even functions in $x$ from $[-K(k), K(k)]$ to $[0,1]$. We extend the domains of $\text{cn}(x,k)$ and $\text{dn}(x,k)$ to the whole of $\mathbb{R}$ by $\text{cn}(x+2K(k),k) = -\text{cn}(x,k)$ and $\text{dn}(x+2K(k),k) = \text{dn}(x,k)$. This implies that $\text{cn}(x,k)$ and $\text{dn}(x,k)$ have $4K(k)$- and $2K(k)$-periodicity. It is shown that $\text{cn}(x,0) = \cos x$, $\text{dn}(x,0) = 1$, $\text{cn}(x,k)$, $\text{dn}(x,k) \to \coth x$ as $k \to 1$ and $\text{cn}(\cdot,k), \text{dn}(\cdot,k) \in C^\infty(\mathbb{R})$.

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
