# Peer review of "Numerical Scheme Based on the Implicit Runge-Kutta Method and Spectral Method for Calculating Nonlinear Hyperbolic Evolution Equations"

_axioms, doi:10.3390/axioms11010028_

Round 1

Reviewer 1 Report

The scientific level of this paper is rather low and the scientific soundness will also be low (it is my opinion). The figures are uncommunicative. The Authors have to explain the Jacobi elliptic functions.

Author Response

Thank you for the comments with recent papers. The modification according to each comments are shown in the attached file.

Reviewer 2 Report

Brief Description of the Work

The work is focused on the construction of a high-precision numerical scheme for solving nonlinear space-time hyperbolic PDEs based on the two-stage and third-order implicit Runge-Kutta method and the spectral method. As an example of application, the authors considered one-dimensional Klein-Gordon equations.

Main Results Obtained

- The constructed scheme has a third-order accuracy with respect to the time;

- The truncation error associated with the spatial variables’ discretisation is much smaller than the error associated with the discretisation of the time variable by setting N (see page 3 for the definition of N) to be sufficiently large;

- The order of computation is O(N log2N).

General Consideration (GC)

GC1) The manuscript is clearly written.

GC2) Please, check the English, several typos have been detected.

GC3) The sentence on page 13, lines 162, is repeated twice.

GC4) There are some points that need to be clarified, some of which are reported below.

Questions/Suggestions (Q/S)

Q/S1) In the numerical solution of space-time hyperbolic partial differential equations, special treatment is required to ensure not only accuracy but also stability of the numerical approximations. The authors scheme guarantees third order accuracy with respect to time, but please also discuss the stability of the solution.

Q/S2) As known, due to the wide variety of hyperbolic PDEs governing physical problems, many hyperbolic PDE-solvers have been developed. For this reason, no particular program is currently recommended as the appropriate solver since it must be selected according to the specific problem that we have to solve. The authors seem to claim that the proposed numerical scheme is applicable in general on the base of two benchmarks. However, we may object that the relations between numerical accuracy and discretisation unit size have been demonstrated by analysing two (quite) smooth cases: the linear case (24) and the non-linear case (26). More concretely, may authors ensure that precision, accuracy, and stability are ensured even when f(u) is a steep function of the variable? or when the system undergoes a bifurcation? The authors are asked to clarify this aspect.

Q/S3) The authors' work is mainly focused on one-dimensional nonlinear hyperbolic evolution equations. However, several relevant physical problems are governed by a class of quasilinear hyperbolic evolution equations including Maxwell's and wave equations over all space or subjected to Dirichlet's or Neumann boundary conditions. Is the authors’ scheme equally applicable for three-dimensional nonlinear hyperbolic evolution equations subjected to Dirichlet's or Neumann boundary conditions?

Q/S4) According to the authors, one of the main advantages of the proposed scheme is that the simultaneous application of the implicit Runge-Kutta method and the spectral method guarantees some quantities remain conserved (contrary to the typical finite difference method which does not provide such assurance). However, recent works show that the application of methods similar to that proposed by the authors such as the pseudo-spectral/collocation methods are equally very effective in solving non-trivial systems of conservation laws. The authors are asked to provide a brief comment about this.

Q/S5) The authors, in addition to showing benchmarks with respect to exact solutions, are also invited to add some comments, or to insert a small Section, where their numerical scheme is compared with other recent developments in the area such as spectral penalty methods, the use of filters, the resolution of the Gibbs phenomenon, and issues related to the solution of nonlinear conservations laws such as conservation and convergence. Please, highlight the added value of the authors' approach with respect to the latter methods. This extra effort will certainly contribute to attract and convince the specialised reader to adopt the proposed numerical scheme.

Conclusions

The work is interesting and well written: it was a pleasure to read it. In my opinion, it deserves to be published. However, there are points that should be clarified, some of which are expressed above. I encourage the authors to take the above suggestions into account; this will certainly contribute to attracting more specialists in the field.

Author Response

Thank you for the comments. The comments are actually valuable for improving the manuscript. All the replies are included in the attached file.

Reviewer 3 Report

My comments are attached.

Author Response

Thank you for a high evaluation of our work. We actually appreciated your evaluation.

Round 2

Reviewer 1 Report

Good paper. I accept your reply. Merry Christmas!

Reviewer 2 Report

The authors answered point-to-point and satisfactorily to all the questions raised in my first report. In my opinion, this new version of the manuscript deserves to be published.